# Extended Venous Resections for Borderline Resectable Pancreatic Head Adenocarcinoma—A Retrospective Studies of Nine Cases

**DOI:** 10.3390/healthcare9080978

**Published:** 2021-07-31

**Authors:** Nicolae Bacalbasa, Irina Balescu, Mihai Dimitriu, Cristian Balalau, Florentina Furtunescu, Florentina Gherghiceanu, Daniel Radavoi, Camelia Diaconu, Ovidiu Stiru, Cornel Savu, Vladislav Brasoveanu, Claudia Stoica, Ioan Cordos

**Affiliations:** 1Department of Visceral Surgery, Center of Excellence in Translational Medicine “Fundeni” Clinical Institute, 022328 Bucharest, Romania; vladislavbrasoveanu@yahoo.ro; 2Department of Obstetrics and Gynecology, Carol Davila University of Medicine and Pharmacy, 050474 Bucharest, Romania; mihaidimitriu@yahoo.ro; 3Department of Surgery, “Ponderas” Academic Hospital, 014142 Bucharest, Romania; irina_balescu206@yahoo.com; 4Department of Obstetrics and Gynecology, “St. Pantelimon” Emergency Hospital, 021659 Bucharest, Romania; 5Department of Surgery, Carol Davila University of Medicine and Pharmacy, 050474 Bucharest, Romania; cristianbalalalu@yahoo.ro; 6Department of Surgery, “St. Pantelimon” Emergency Hospital, 021659 Bucharest, Romania; 7Department of Public Health and Management, University of Medicine and Pharmacy “Carol Davila”, 050474 Bucharest, Romania; florentinafurtunescu@yahoo.ro; 8Department of Marketing and Medical Technology, “Carol Davila” University of Medicine and Pharmacy, 050474 Bucharest, Romania; florentinagherghiceanu@yahoo.ro; 9Department of Urology, ‘Prof. Dr. Th. Burghele’ Clinical Hospital, 061344 Bucharest, Romania; danielradavoi@yahoo.ro; 10Department of Urology, “Carol Davila” University of Medicine and Pharmacy, 050474 Bucharest, Romania; 11Department of Internal Medicine, “Carol Davila” University of Medicine and Pharmacy, 050474 Bucharest, Romania; cameliadiaconu@yahoo.ro; 12Department of Internal Medicine, Clinical Emergency Hospital of Bucharest, 014461 Bucharest, Romania; 13Emergency Institute for Cardiovascular Diseases Prof. Dr. C.C. Iliescu, 022328 Bucharest, Romania; ovidiustiru@yahoo.ro; 14Department of Cardio-Thoracic Pathology, “Carol Davila” University of Medicine and Pharmacy, 050474 Bucharest, Romania; 15Department of Thoracic Surgery, “Marius Nasta” National Institute of Pneumology, 050159 Bucharest, Romania; cornelsavu@yahoo.ro (C.S.); ioancordos@yahoo.ro (I.C.); 16Department of Thoracic Surgery, “Carol Davila” University of Medicine and Pharmacy, 050474 Bucharest, Romania; 17Department of Surgery, Ilfov County Hospital, 077160 Bucharest, Romania; claudiastoica@yahoo.ro; 18Department of Anatomy, Carol Davila University of Medicine and Pharmacy, 050474 Bucharest, Romania

**Keywords:** pancreatic cancer, borderline, resection, venous invasion, reconstruction

## Abstract

Background: pancreatic cancer is one of the most lethal malignancies and a leading cause of cancer-related death worldwide. The only chance to improve the long-term outcomes of patients with pancreatic cancer is surgery with radical intent. Methods: in the present paper, we aim to describe a case series of 9 patients submitted to radical surgery for borderline resectable pancreatic cancer. Results: in all cases, negative resection margins were achieved. The types of venous resection consisted of tangential portal vein resection in four cases, circumferential portal vein resection with direct reanastomosis in one case and circumferential resection with graft placement in another four cases; postoperatively, one patient developed a vascular surgery-related complication consisting of graft thrombosis and thus necessitated prolonged anticoagulant therapy. Conclusions: extended venous resections can be a safe and efficient way to maximize the benefits of radical surgery in locally advanced, borderline resectable pancreatic cancer.

## 1. Introduction

Pancreatic adenocarcinoma is a particularly aggressive malignancy which represents a leading cause of cancer-related death worldwide. Therefore, the lifespan in such cases remains extremely poor, usually being only a few months following the time of the initial diagnosis [1,2]. The outcomes of these cases significantly improved after performing radical surgical procedures such as pancreatoduodenectomy, and the benefits in terms of survival rate were quickly demonstrated in spite of the fact that significant perioperative complications might develop [3,4,5,6]. However, at the time of the initial diagnosis, less than 20% of cases present resectable lesions, as local invasions or distant metastases are frequently encountered [7,8]. When it comes to locally advanced pancreatic head carcinoma, it has been considered that local invasion is the sign of a more aggressive biology of the tumor; however, once the vascular techniques have been widely implemented and extended vascular resections have become more commonly performed, the long-term outcomes came to demonstrate improved rates of survival. It also demonstrated that the theory of biologically aggressive tumors should not be taken into further consideration [9,10,11,12]. The benefits have been most widely reported in cases in which a limited invasion of the portal system is encountered and considered as a borderline resectable lesion. Meanwhile, cases presenting both venous and arterial invasion should be carefully analyzed; the decision between surgery and neoadjuvant chemotherapy followed by surgery is a very difficult one [13,14,15]. The aim of this paper is to report a series of nine patients submitted to surgery for borderline resectable pancreatic head adenocarcinoma presenting limited invasion at the level of the portal system.

## 2. Materials and Methods

After obtaining the approval of the Ethics Committee of Fundeni Clinical Institute no. 191/2021, data of patients submitted to venous resections en bloc with pancreatic head resections were retrospectively reviewed; among the 12 identified cases, 3 patients needed arterial resections and were thus excluded from the study. Finally, nine cases were considered as eligible for this study. All patients had been submitted to surgery in Fundeni Clinical Institute between January 2020 and April 2021.

## 3. Results

The mean age at the time of surgery was of 53 years (range 43–62 years) while the sex ratio male/female was 7/2. In all cases, the serum values of CA19-9 were preoperatively measured and the mean value was 312 U/mL (range 23–528 U/mL); interestingly, in one case a non-secretant tumor was found, the serum level of CA 19-9 being 23 U/mL while the biopsy demonstrated the presence of a moderately differentiated pancreatic adenocarcinoma. All cases were preoperatively submitted to computed tomography, in two out of the nine cases, a venous invasion was not described; meanwhile, in all of the cases magnetic resonance imaging was performed in order to better evaluate the local status as well as of the possibility of association with distant metastases. Meanwhile, in all of the cases an endoscopic ultrasound was performed, and a biopsy was retrieved; at the time, a local venous invasion was objected in eight out of the nine cases. As for the result of the histopathological studies of the biopsies, in all of the cases the presence of pancreatic ductal adenocarcinoma was observed. The most significant associated comorbidities were diabetes mellitus in five cases, obesity in three cases, arterial hypertension in three cases and chronic pulmonary obstructive disease in two cases.

After the final preoperative workup, all of the cases were submitted to surgery with curative intent. In all of the cases, pancreatoduodenectomy was associated with portal vein resection. Intraoperative details are shown in Table 1.

Postoperatively, a single patient developed a complication related to vascular resection and consisted of a partial graft thrombosis which necessitated prolonged anticoagulant treatment with low-molecular heparin for the next month, followed by oral anticoagulant for the next two months. At the third month, a follow-up of all of the cases showed a satisfactory venous blood flow at the level of the portal system.

## 4. Discussion

The subject of vascular resections as part of radical surgery for locally advanced pancreatic cancer has been widely debated in the last decades, and the most appropriate therapeutic strategy has seen permanent changes, being significantly influenced by the development in the field of surgical technique and perioperative management of such cases [16,17,18,19]. Initially suggested by Moore et al. in 1951 and Fortner et al. in 1973, the procedure was considered at that moment to be too aggressive, associated with increased rates of perioperative complications; therefore, at that time, most patients submitted to such radical surgical procedure ended up dying in the early postoperative period and the long-term benefits could not be further evaluated [20,21].

However, the fact that the notion of portal vein resection is a very generic one which includes tangential, short circumferential or extended circumferential resections of the portal system should not be omitted. Therefore, according to the extent of the local invasion, several classification systems have been proposed. The most commonly used is that proposed by the International Study Group of Pancreatic Surgery in 2014; according to this system, tangential resections were classified as type I if per primam venous suture is possible or type II if graft is needed, respectively, type III and IV if circumferential resection is performed. Cases in which an end-to-end anastomosis was feasible were considered as type III resections while cases in which the length of the resected segment did not allow an end-to-end anastomosis and in which a graft was needed were classified as type IV resections [22]. As expected, patients included in the first two categories usually present less extended lesions, necessitate a less aggressive surgical procedure and are expected to have a more favorable long-term outcome compared to the cases in the third and fourth group.

Meanwhile, another important subject which has been debated is related to the extent of the pancreatic resection whenever vascular resections are needed. Therefore, certain studies raised the question of whether total pancreatectomy should become a standard procedure whenever vascular resections are needed, in order to minimize the risk of perioperative complications and maximize the rates of negative resection margins [23]. One of the most reluctant studies conducted on this issue was published by Serenari et al. in 2019 and included 99 patients submitted to pancreatic and venous resections for borderline resectable pancreatic cancer [24]. Among these cases, there were 25 patients submitted to pancreatoduodenectomy and tangential portal vein resection, 12 cases submitted to pancreatoduodenectomy and segmental portal vein resection, 23 cases submitted to total pancreatectomy and tangential portal vein resection and 39 patients submitted to total pancreatectomy and segmental portal vein resection. The authors demonstrated that there were no significant differences in terms of perioperative complications (at 30 day and 90 day follow-up) between the four groups; however, when it came to the long-term outcomes, the median overall survival of patients submitted to total pancreatectomy and segmental portal vein resection was significantly poorer compared to those submitted to pancreatoduodenectomy and tangential portal vein resection (7, 9 months versus 29, 5 months, *p* = 0.001). Meanwhile, there were no significant differences between cases submitted to pancreatoduodenectomy and segmental portal vein resections and those submitted to total pancreatectomy and tangential venous resection. Furthermore, the multivariate analysis demonstrated that the necessity of total pancreatectomy as well as the necessity of performing a circumferential resection were the only independent poor prognostic factors affecting the overall survival of the patients. This fact was explained through the fact that cases necessitating segmental resection or total pancreatectomy most frequently presented larger tumors or more advanced disease and a higher risk of microscopically positive resection margins (defined as a R1 resection) [24].

Another very important issue which should be taken in consideration in cases where portal vein invasion is present is the location of the invaded segment. Therefore, Chaoyang et al. consider this the greatest inconvenience regarding the classification proposed by the International Study Group of Pancreatic Surgery in 2014 and therefore have proposed a new classification system [25]. According to this new system, cases in which tangential portal vein resection, followed by direct vein suturing, are classified as type I cases; those in which circumferentialportal vein invasion is present and in which portal resection followed by end-to-end anastomosis is feasible are classified as type II resections; cases in which circumferential portal vein resection followed by graft placement is performed are classified as type III resections; cases in which the venous resection is extended to the superior mesenteric vein and in which a graft anastomosis with the jejunal veins is needed is classified as type IV resection. Meanwhile, the authors underline the benefits of reimplantation the mesenterico-lienal confluent at the level of the graft in cases submitted to type III or IV resections in order to prevent the development of further complications such as gastric varices or splenic infarctation. The authors conducted a study on 11 patients submitted to type I resection, 15 to type II resection, 18 to type III resection and eight cases to type IV resection; they underlined the fact that patients submitted to type III and IV resections reported similar operative times and estimated blood loss amounts, which were significantly higher when compared to those included in the first two categories. Meanwhile, the long-term outcomes demonstrated that the longest median overall survival time was reported after type I resection. Furthermore, cases submitted to type II and III resections reported similar outcomes which were significantly improved when compared to cases submitted to type IV resection. Therefore, this system seems to provide a more efficient stratification of patients in whom venous resection is tempted regarding both short-term and long-term outcomes [25].

## 5. Conclusions

Although venous invasion in pancreatic head adenocarcinoma was initially considered as the formal contraindication for surgery, recently published results demonstrate that, in selected cases, extended resections of the pancreatic tumor en block with the invaded segment of the venous structure can be safely performed in selected cases with acceptable postoperative morbidity rates. Meanwhile, the long-term survival outcomes in such cases seem to be similar to the ones reported in cases submitted to standard pancreatic resections, once again demonstrating the efficacy of the method.

## Figures and Tables

**Table 1 healthcare-09-00978-t001:** Intraoperative details of patients submitted to pancreatic head resection en bloc with portal vein resection.

Case	Sex	Degree of Differentiation	Type of VascularResection	Type of Reconstruction	Estimated Blood Loss (ml)	Length of Surgery (min)	Type of Resection
1	M	Moderately differentiated	Circumferential portal vein resection	Graft placement	550	480	R0
2	M	Well differentiated	Tangential portal vein resection	Phleborrhaphy	450	300	R0
3	M	Well differentiated	Tangential portal vein resection	Phleborrhaphy	600	350	R0
4	F	Well differentiated	Tangential portal vein resection	Phleborrhaphy	550	420	R0
5	M	Moderately differentiated	Tangential portal vein resection	Phleborrhaphy	400	400	
6	M	Poorly differentiated	Circumferential portal vein resection extended to superior mesenteric vein resection	Graft placement	1000	660	R0
7	F	Moderately differentiated	Circumferential portal vein resection	Direct end to end anastomosis	500	480	R0
8	M	Poorly differentiated	Circumferential portal vein resection extended to superior mesenteric vein resection	Graft placement	1100	550	R0
9	M	Moderately differentiated	Circumferential portal vein resection extended to superior mesenteric vein resection	Graft placement	800	500	R0

## Data Availability

Data supporting reported results are available from the corresponding author at request.

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
