# Peer review of "Extended Venous Resections for Borderline Resectable Pancreatic Head Adenocarcinoma—A Retrospective Studies of Nine Cases"

_healthcare, 2021, doi:10.3390/healthcare9080978_

Round 1
Reviewer 1 Report
This is an interesting work. However, the paper is not presented well. English statement such as "... destroy the hypothesis ..." in the introduction is something unconventional. The phrase "A case series" in the title should be revised to "A retrospective study of nine cases". The conclusion paragraph is not complete. The references and in-text quotations are confusing.
My specific comments are below:
- The title should be revised to "Extended venous resections for borderline resectable pancreatic head adenocarcinoma – a retrospective studies of nine cases".
- Reference numbers should be added on page 4 line 139 after Serenari et al. and line 163 after Chaoyang et al. for consistency of reference citations.
- Re-write the conclusions. It is in-complete at present form.
- English expressions should be checked by a native speaker, such as, "...destroyed the hypothesis..." on page 2 line of 71-72 is un-conventional to me.
Author Response
We have performed all the demanded modifications:
- we modified the title and the conclusions
- we modified the text where necessary
- we added the references
Thank you for your time to review our paper!

Reviewer 2 Report
The manuscript reports on 9 cases admitted to surgery for borderline resectable pancreatic head adenocarcinoma.
The following can be added to the manuscript:
- the time range when and the centre where the surgeries were performed
- if available data on patient comorbidities at the surgery time
- complete and end the conclusion
Author Response
We have performed the demanded modifications:
- we modified the conclusions
- we described patients' comorbidities
- we gave details abou time and place of surgery
- Thank yoy for revieweing our paper!
